# Assessment of the Effective Impact of Bisphenols on Mitochondrial Activity, Viability and Steroidogenesis in a Dose-Dependency in Human Adrenocortical Carcinoma Cells

**Nikola Knížatová** [1,*] , **Hana Greifová** [1] , **Katarína Tokárová** [1] , **Tomáš Jambor** [2] , **Łukasz J. Binkowski** [3] **and Norbert Lukáč** [1]

1   Department of Animal Physiology, Faculty of Biotechnology and Food Sciences, Slovak University of Agriculture in Nitra, Tr. A. Hlinku 2, 949 76 Nitra, Slovakia; hana.greifova@uniag.sk (H.G.); katarina.tokarova@uniag.sk (K.T.); norbert.lukac@uniag.sk (N.L.)
2   BioFood Centre, Faculty of Biotechnology and Food Sciences, Slovak University of Agriculture in Nitra, Tr. A. Hlinku 2, 949 76 Nitra, Slovakia; tomas.jambor@uniag.sk
3   Faculty of Exact and Natural Sciences, Institute of Biology, Pedagogical University of Cracow, Podchorążych 2, 30-084 Kraków, Poland; lukasz.binkowski@up.krakow.pl
*   Correspondence: nikola.knizatova@gmail.com or xknizatovan@uniag.sk; Tel.: +421-37-641-4288

**Abstract:** In recent years, bisphenol analogues such as bisphenol B (BPB), bisphenol F (BPF), and bisphenol S (BPS) have come to replace bisphenol A (BPA) in food packaging and food containers, since BPA has been shown to leach into food and water, causing numerous negative health effects. Although much information on the endocrine activity of BPA is available, a proper human hazard assessment of analogues that are believed to have a less harmful toxicity profile is lacking. The aim of our in vitro study was to assess the potential effect of bisphenol B, F, and S on the biosynthesis of steroid hormones in human H295R adrenocortical carcinoma cells, using the enzyme-linked immunosorbent assay. In addition, we evaluated mitochondrial activity using the MTT test and viability using triple assay. Adrenocortical carcinoma cells were cultivated for 24 h in the presence of bisphenol B, F, or S (0.1, 0.5, 1, 10, 25, 50, 75, 100 μM). We demonstrated that BPB, BPF, and BPS could affect progesterone and testosterone secretion, as well as affect cell mitochondrial, lysosomal, and metabolic activity, as well as plasma membrane integrity, but considerably more detailed and systematic research is required for a better understanding of risks associated with the effects of bisphenols on steroidogenesis.

**Keywords:** adrenocortical carcinoma cells; H295R; viability; mitochondrial activity; steroidogenesis; testosterone; progesterone

## 1. Introduction

The growing global production and release of industrial chemicals into the environment have prompted scientists to speculate that current pollutants such as bisphenols may unquestionably disrupt health conditions, resulting in extensive physiological function damage via endocrine disruption [1,2]. Bisphenols (BPs) have been used all over the world for decades. Bisphenol compounds can be found in plastics used by consumers for food storage, baby formula packaging, baby bottles, the lining of canned food and drink cans, dental implants, and sales receipts [3].

The first indications about the possible leakage of bisphenol monomers into food and drink were published in 2007. BPA (bisphenol A) has since become one of the most well-known EDCs (endocrine-disrupting chemicals) because of its strong effects on steroid hormone production [2,4–8], disturbed mammary gland development [9], changes in obesity-associated parameters [10], reproductive and developmental toxicity [11], heart disease, diabetes, abnormal liver function [12–14], and the nervous system [15,16]. The significant majority of the population is exposed to BPA, according to Zhang et al. (2011)

and Calafat et al. (2008) [5,17]. Diet is estimated as the main source of human exposure, followed by thermal paper [18]. Generally speaking, humans consume less than 1 g/kg body weight per day of BPA, although occupationally exposed individuals had considerably greater BPA levels in the blood than those exposed to BPA in the environment [19,20]. According to studies from numerous countries and on people of various ages, the quantity of BPA in human blood varies from 0.2 to 20 ng/mL (0.0008–0.088 μM) [12]. BPA levels in the urine of Chinese males working in facilities making semiautomatic epoxy resin have been reported to reach 1934.85 ng/mL, which corresponds to experimental dosages of 8.41 μM [21]. Heinälä et al. (2017) report BPA levels reaching 1273 ng/mL in workers in Finland working at plants manufacturing thermal paper. This value corresponds to experimental dosages of 5.55 μM [22].

As BPA is being removed from consumer items, there is a progressive move to its analogues, bisphenols B, F, and S, as polycarbonate resin components are occurring [23]. The industry quickly formulated new BPA-free plastics, which were made using bisphenol analogs with highly similar structure and chemical properties to be used in packing and storage containers for food instead of BPA as a result of a ban on BPA-containing products for babies in the EU, and guidelines to avoid using BPA for baby bottles and formula packaging in the US. Bisphenols are fabricated by combining phenol with acetone (BPA), 2−butanone (BPB), formaldehyde (BPF), or sulfur trioxide (BPS). The recent monomer, BPS, is the most common BPA analog marketed in BPA-free products [24–26]. BPB, BPF, and BFS can be found in canned soft beverages and foods, as well as thermal receipt paper [27–30]. BPA has the highest concentration in food items, followed by BPF, BPS, and BPB [30]

Previous research has found that direct inhibition of BPA analogs on steroidogenesis and hormonal imbalance is extensively discussed. Additionally, BPA substitutes have been shown to have neurotoxicity, genotoxicity, reproductive toxicity, and significant endocrine-disrupting effects, according to several studies [2,30–33]. Furthermore, BPS has been found in human urine samples [34]. Eladak et al. (2015) observed that rat and mouse fetal testes are at least 100 times less sensitive to BPA than human fetal testes. They demonstrated that in basal conditions, BPS and BPF, which are gradually replacing BPA, have antiandrogenic effects similar to the antiandrogenic properties that are comparable to BPA's [35].

Sexual steroid hormones are important regulators of reproduction in vertebrates, and they also play a role in a variety of other developmental and growth processes. As a result, substances that alter steroid hormone synthesis may be directly related to adverse outcomes for these processes [36,37]. Toxicological data are currently limited, and experimental research evaluating BPA analogue effects is unclear. As a result, we decided to investigate the effects of BPB, BPS, and BPF on cellular toxicity and possible steroidogenesis-disrupting activity in vitro.

Cell lines are an excellent biological model for investigating the direct impact of various chemical and physical variables on steroidogenesis, H295R cells are a useful tool for detecting hazardous substances that interfere with steroidogenesis [38–40]. The test guideline of the H295R steroidogenesis assay (TG 456) was verified by the Organization for Economic Cooperation and Development [41]. The human adrenocortical cancer cell line NCI-H295R was used as a model system for detecting the effects of BPB, BPS, and BPF on the secretion of sex steroid hormones (testosterone, progesterone) in vitro in the current study. H295 cells used to establish this cell line were obtained from a primary hormonally active adrenocortical carcinoma [38,39].

The objective of our study was to determine the effects of bisphenol B, F, and S on the steroidogenesis of the human adrenocortical carcinoma cell line (NCI-H295R). Specifically, we examined the dose-dependent changes of bisphenols as endocrine disruptors concerning the production of progesterone and testosterone by adrenocortical carcinoma cells in vitro. Quantification of steroid hormones was performed using the enzyme-linked immunosorbent assay (ELISA). In addition, we evaluated mitochondrial activity (using the MTT test) and cell viability parameters (using the triple test).



## 2. Materials and Methods

### 2.1. H295R Cell Culture and Treatment

The American Type Culture Collections provided the NCI-H295R cells (ATCC CRL-2128; ATCC, Manassas, VA, USA). The cells were cultured using protocols that had previously been established and validated. After starting the H295R culture from the ATCC batch, the cells were cultured for four passages before being split and frozen in liquid nitrogen. To obtain optimum hormone synthesis, the cells used in the following experiments were cultured for a minimum of three more passages. The H295R cells were grown in 25 cm$^2$ plastic tissue culture flasks (TPP, Trasadingen, Switzerland) in Dulbecco's Modified Eagle Medium/Nutrient F-12 Ham 1:1 mixture (Sigma, St. Louis, MO, USA) supplemented with 1.2 g/L NaHCO$_3$ (Molar Chemicals Halasztelek, Hungary), 12.5 mL/L of BD Nu-Serum (BD Bioscience, Bath, UK), and 5 mL/L of ITSC Premix (Corning, AZ, USA) in a CO$_2$ incubator at 37 °C with a 5% CO$_2$ atmosphere. The culture medium was changed four times per week, and it was removed from the culture flasks once an acceptable cell density had been achieved. With 0.25% trypsin-EDTA (Sigma-Aldrich, St. Louis, MO, USA) for 4 min, the H295R cells were detached from the bottom of the 25cm$^2$ culture flasks. The cells were then centrifuged for 5 min at 125× *g* before being resuspended in a fresh cell culture medium. A hemocytometer (Burker chamber) was used to count the cells and adjust the concentration to the required level. The cell suspension was plated into sterile 96-well cell culture plates (60,000 cells/100 μL/well) for cytotoxicity and hormone measurements. The cells were incubated for 24 h in a CO$_2$ incubator at 37 °C under a humidified atmosphere of 95% air and 5% CO$_2$. To explore the effect of bisphenols, cells were cultured for 24 h in a medium containing specific concentrations of each bisphenol (0.1, 0.5, 1, 10, 25, 50, 75, 100 μM; Sigma-Aldrich, St. Louis, MO, USA). Cells without any treatment served as a control group; DMSO (0.1%) served as a negative control. The specific concentration range of bisphenols was chosen based on the findings of our pilot range-finding studies. Each experiment was repeated 3 times with cells from different passages each time.

### 2.2. Mitochondrial Activity Assay

The MTT (3-4,5-dimetyltiazol-2-yl)-2,5-diphenyltetrazolium bromide) (NR; Sigma-Aldrich, St. Louis, MO, USA) the test was used to evaluate the mitochondrial activity of adrenocortical carcinoma cells exposed to various concentrations of bisphenol B, F, and S. The reduction in a yellow tetrazolium salt to insoluble blue formazan in the mitochondria of live cells was measured in this assay [42]. After 24 h of treatment, cells were incubated in a CO$_2$ incubator for 1 h with MTT tetrazolium salt (Sigma-Aldrich, St. Louis, MO, USA). The supernatants were then removed, and the formed formazan crystals were dissolved with isopropanol (p.a. CentralChem, Bratislava, Slovak Republic). An ELISA reader (Multi-scan FC, ThermoFisher Scientific, Vantaa, Finland) was used to measure dissolved formazan at 570 nm against 620 nm wavelengths. All of the data were expressed as a percentage of the control group.

### 2.3. Triple Assay

Three cellular activities were monitored for cell viability using three different indicator dyes were used: metabolic activity with alamarBlue (ThermoFisher Scientific, Waltham, MA, USA), plasma membrane integrity with 5-carboxyfluorescein diacetate acetoxymethyl ester (CFDA-AM; ThermoFisher Scientific, Waltham, MA, USA), and lysosomal activity with neutral red (NR; Sigma-Aldrich, Waltham, MA, USA). The resazurin reduction test is also known as the AlamarBlue assay. Resazurin is a nontoxic, cell-permeable, blue nonfluorescent redox indicator that may be used to measure the number of live cells by measuring the quantity of resorufin-to-resorufin conversions by mitochondrial and other enzymes, such as diaphorases. When resazurin enters cells, it is converted to resorufin, a red, highly fluorescent color. Viable cells convert resazurin to resofurin continually, enhancing the fluorescence of the culture medium. A microplate reader fluorometer can be used to measure the ratio of cell metabolic activity. Another fluorogenic dye that indicates plasma

membrane integrity is CFDA-AM. It is a nontoxic esterase substrate that can be metabolized by nonspecific esterases in live cells from a membrane-permeable, nonpolar, nonfluorescent material to polar, fluorescent carboxyfluorescein (CF). This conversion indicates the plasma membrane's integrity, as only an intact membrane can sustain the cytoplasmic milieu required for esterase activity. Neutral red staining belongs to the colorimetric assays. Nonionic passive diffusion allows this weakly cationic dye to penetrate cell membranes and concentrate in lysosomes. The dye is then released from the viable cells using an acidified ethanol solution, and the dye's absorbance is spectrophotometrically measured.

Schirmer et al. (1997) were the first to describe the use of these three dyes to provide an overview of the cytotoxicity/cytoprotectivity of treatments to cells in 96-well plates [43]. This protocol was followed in this experiment with minimal changes. In summary, our method measured three cell viability parameters on the same set of cells at the same time without interfering. After a 24 h treatment, cells seeded in 96-well plates were treated with a solution of almarBlue and CFDA-AM in Eagle's minimal essential medium (Sigma-Aldrich, St. Louis, MO, USA). After an hour of incubation, the cells were measured at different wavelengths. The cells were then rinsed in PBS and exposed to neutral red dye in the minimal essential medium for 1 h. After incubation, the cells were rinsed twice with PBS (phosphate-buffered saline; Sigma-Aldrich, St. Louis, MO, USA) and exposed to the lysis buffer for 30 min before being measured at a specific wavelength. The multiple endpoint assay is based on measurements performed at 525/580–640 nm for alamarBlue, 490/510–570 nm for CFDA-AM, and 525/660–720 nm for NR with a Glomax Multi + Combined Spectro-Fluoro Luminometer (Promega Corp., Madison, WI, USA). The data were collected from three different experiments and expressed as a percentage of the control group, which was set at 100%.

### 2.4. Evaluation of Testosterone and Progesterone Production

The media were collected after a 24 h culture in the presence of various bisphenol concentrations and centrifuged at $300\times g$ for 10 min at 4 °C. The supernatant was stored at a temperature of −20 °C. To quantify testosterone and progesterone directly from aliquots of the culture medium, an enzyme-linked immunosorbent assay (ELISA) was performed. The concentration of progesterone in the culture media was determined using an enzyme-linked immunosorbent assay (ELISA) kit (Cat. #K00225, Dialab, Wiener Neudorf, Australia), according to the manufacturer's instructions. ELISA kits (Cat. #K00234, Dialab, Wiener Neudorf, Austria) were used to measure testosterone concentrations. An ELISA reader was used to measure the absorbance at 450 nm (Mul-ti-scan FC, ThermoFisher Scientific, Vantaa, Finland).

### 2.5. Statistical Analysis

The data gathered were statistically analyzed using GraphPad Prism 8 (GraphPad Software Incorporated, San Diego, CA, USA). The initial step was to evaluate descriptive statistical parameters (minimum, maximum, and standard error). One-way analysis of variance (ANOVA) and Dunnett's multiple comparison test were used to examine differences between bisphenol treatments and the control. The significance levels were established at *** ($p < 0.001$), ** ($p < 0.01$), and * ($p < 0.05$). Data were obtained from three sets of independent experiments ($n = 3$). The results were presented as means (SEM) of mitochondrial activity percent, metabolic activity percent, membrane integrity percent, lysosomal activity percent, progesterone percent, and testosterone percent of a control group.

## 3. Results

### 3.1. Mitochondrial Activity

Our findings show that bisphenols may act dose-dependently as a stimulant at low doses or as an inhibitor at high levels, based on the MTT test. In the case of cells treated with BPB, as shown in Figure 1, we observed slightly increased mitochondrial activity in the case of cells treated with 0.1 μM BPB ($p > 0.05$). Significantly lower ($p < 0.01$) values

were observed in the experimental groups with the addition of 50, 75, and 100 μM BPB in comparison with a control group. Similar results were obtained in the case of BPF, but with more pronounced differences. The lowest concentrations of BPF caused stimulation of H295R cells' mitochondrial activity without significant changes ($p > 0.05$), but higher doses (1–100 μM) of BPF caused inhibition of mitochondrial activity ($p < 0.001$). As seen in Figure 1, 0.1 and 0.5 μM BPS caused a nonsignificant increase in mitochondrial activity. On the contrary, 1, 50, and 100 μM BPS after 24 h of culture inhibited cell mitochondrial activity, resulting in a significant decrease in values ($p < 0.05$).

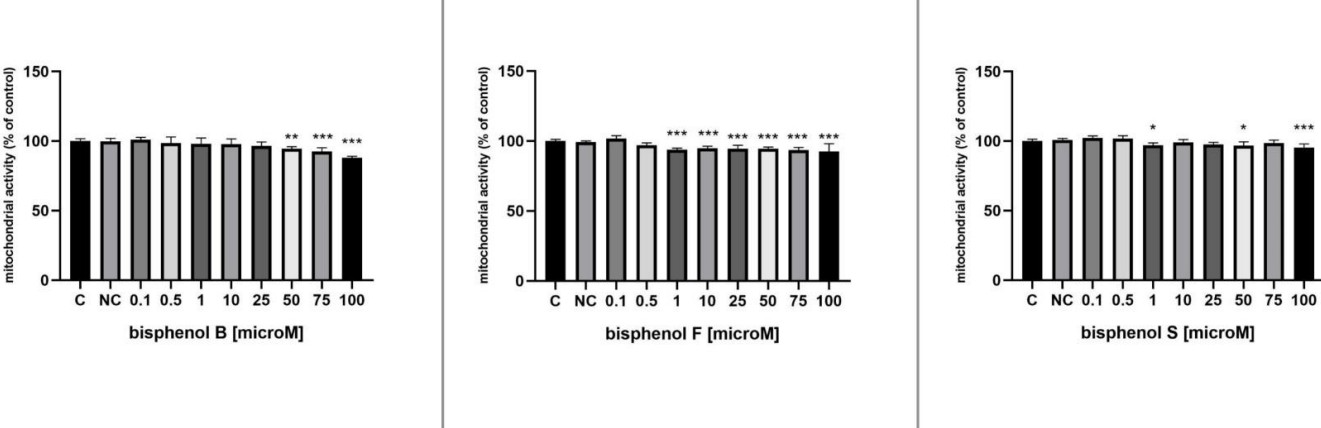

**Figure 1.** The impact of bisphenol B, F, and S on H295R cells' mitochondrial activity in vitro. Abbreviations: C—control group, NC—negative control. The mean (±SEM) optical density percent of the control (untreated) and treated (bisphenol) groups is represented by each bar. The data were collected from three separate experiments. Between the control and experimental groups, the significance levels were established at *** ($p < 0.001$), ** ($p < 0.01$), and * ($p < 0.05$).

*3.2. Metabolic Activity*

Bisphenols may function dose-dependently as a stimulant at low concentrations or as an inhibitor at high concentrations, according to an analysis of metabolic activity using alamarBlue. As seen in Figure 2, in the case of cells treated with BPB, the lowest concentration (0.1 μM) led to an increase in metabolic activity without significant changes ($p > 0.05$). Higher doses of BPB, on the other hand, inhibited cell metabolic activity after 24 h of culture, with a significant drop in values ($p > 0.001$) for experimental groups supplemented with 1–100 μM, compared with an untreated control group. Similar results were obtained in the case of BPF. The lowest concentrations of BPF caused stimulation of H295R cells' metabolic activity (0.1 μM; $p < 0.05$), but higher doses (10–100 μM) of BPF caused inhibition of metabolic activity ($p < 0.01$). The metabolic activity of cells cultured with BPS showed a similar trend in the concerning experiment, where H295R cells were exposed to BPS for 24 h, although with less pronounced differences than in the previous cases. Lower concentrations of BPS (0.1 and 0.5 μM) caused stimulation of H295R cells' mitochondrial activity ($p > 0.05$). Significantly decreased ($p < 0.01$) metabolic activity was recorded only in the experimental group treated with the highest dose of BPS (100 μM) ($p < 0.01$).

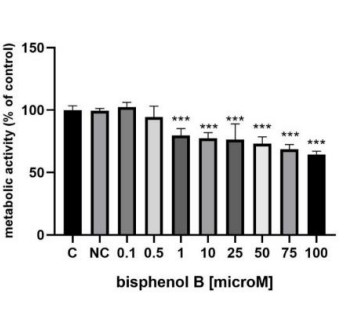
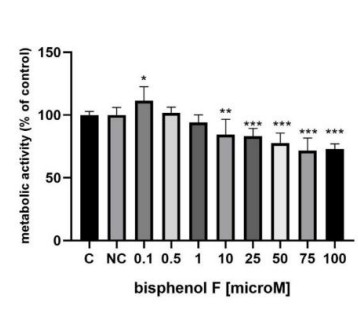
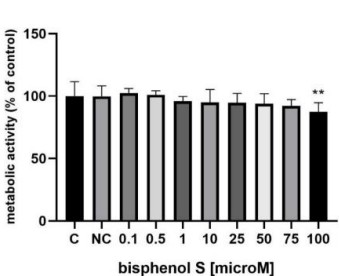

**Figure 2.** The impact of bisphenol B, F, and S on H295R cells metabolic activity in vitro. Abbreviations: C—control group, NC—negative control. The mean (±SEM) optical density percent of the control (untreated) and treated (bisphenol) groups is represented by each bar. The data were collected from three separate experiments. Between the control and experimental groups, the significance levels were established at *** ($p < 0.001$), ** ($p < 0.01$), and * ($p < 0.05$).

### 3.3. Membrane Integrity

The effect of bisphenols on the integrity of the H295R cells cell membrane is presented in Figure 3. Bisphenols have a biphasic effect on membrane integrity, similar to previous experiments (mitochondrial activity, metabolic activity). Cell membrane integrity measurement in H295R cells treated with BPB for 24 h showed an increase in the experimental group treated with 0.1 μM ($p > 0.05$) and decrease in experimental groups treated with 25, 50, 75, and 100 μM ($p < 0.05$). Only in experimental groups supplemented with higher doses of BPF (25, 50, 75, and 100 μM), which resulted in a decline in values, did cell membrane integrity measurements in H295R cells treated with BPF for 24 h indicate significant alterations ($p < 0.01$) (Figure 3); on the contrary, we observed a nonsignificant increase in experimental groups treated with a lower concentration of BPF (0.1 and 0.5 μM). We observed the strongest substantial impact on membrane integrity in bisphenol S-treated groups. Significant changes ($p < 0.01$) were recorded in groups treated with 0.5, 1, 10, 25, 50, 75, and 100 μM BPS—we observed a significant decline in membrane integrity. The lowest concentrations of BPS (0.1 μM) caused non-significantly increased cell membrane integrity in H295R cells in comparison with the control group.

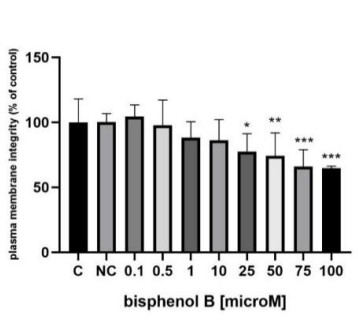
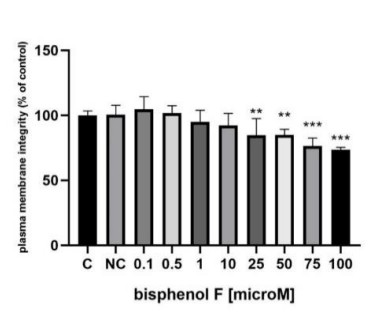
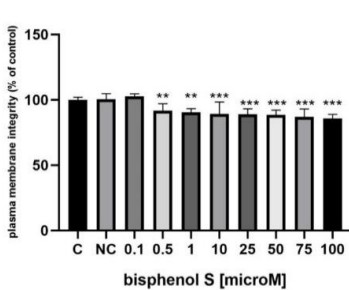

**Figure 3.** The impact of bisphenol B, F, and S on H295R cells membrane integrity in vitro. Abbreviations: C—control group, NC—negative control. The mean (±SEM) optical density percent of the control (untreated) and treated (bisphenol) groups is represented by each bar. The data were collected from three separate experiments. Between the control and experimental groups, the significance levels were established at *** ($p < 0.001$), ** ($p < 0.01$), and * ($p < 0.05$).

### 3.4. Lysosomal Activity

Figure 4 shows the effect of bisphenols on lysosomal activity in H295R cells. BPB at 0.1 μM resulted in a non-significant ($p > 0.05$) elevation of lysosomal function compared with the control group, but higher dosages (25, 50, 75, and 100 μM) resulted in a significant ($p < 0.001$) decrease in lysosomal activity. In the case of BPF, significant changes ($p < 0.05$) were recorded only in experimental groups of cells cultivated in the presence of higher concentrations (25, 50, 75, and 100 μM). Lower concentrations of BPF (0.1, 0.5, 1 μM) improved lysosomal activity of cells non-significantly ($p > 0.05$) compared with the control group. Similarly, experimental groups treated with 0.1 ($p > 0.05$), 0.5 ($p < 0.001$), and 1 M ($p > 0.05$) of BPS showed an increase in lysosomal activity. Experimental groups of cells cultivated in the presence of higher concentrations (25, 50, 75, and 100 μM) of BPF showed significantly lower ($p < 0.01$) lysosomal activity when compared against the control group.

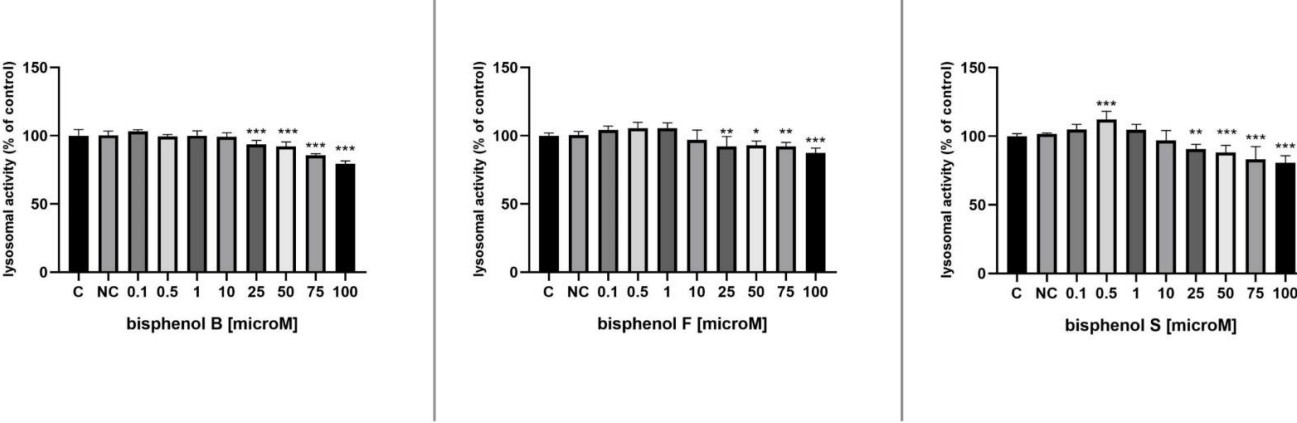

**Figure 4.** The impact of bisphenol B, F, and S on H295R cells lysosomal activity in vitro. Abbreviations: C—control group, NC—negative control. The mean (±SEM) optical density percent of the control (untreated) and treated (bisphenol) groups is represented by each bar. The data were collected from three separate experiments. Between the control and experimental groups, the significance levels were established at *** ($p < 0.001$), ** ($p < 0.01$), and * ($p < 0.05$).

### 3.5. Progesterone Secretion

The impact of bisphenols on progesterone secretion in vitro is shown in Figure 5. A total of 24 h of H295R cells cultivation with BPB showed significant changes with increased production of progesterone in samples supplemented with 0.1 and 1 μM BPB ($p < 0.01$) and non-significant increase in samples supplemented with 0.5 μM, on the other hand, higher concentrations of BPB (25, 50, 75, and 100 μM) led to significant ($p < 0.001$) reduction in progesterone biosynthesis in treated cells. After BPF was added to the culture media, it resulted in significantly higher ($p < 0.05$) production of progesterone in experimental groups supplemented with 0.1–75 μM and significantly lower ($p < 0.01$) production of progesterone in the experimental group supplemented with 100 μM. Progesterone levels were also significantly ($p < 0.01$) elevated, as seen in Figure 5, in the presence of 0.1 μM of BPS and higher concentrations of BPS (25–100 μM) led to a significant ($p < 0.001$) reduction in progesterone biosynthesis in cells exposed to BPS; these results are similar to results obtained after cultivation with BPB.

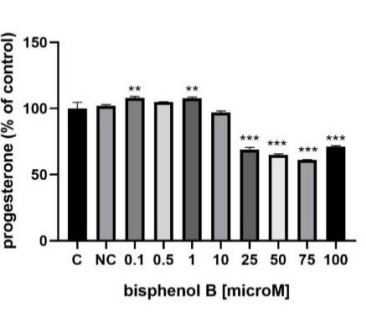
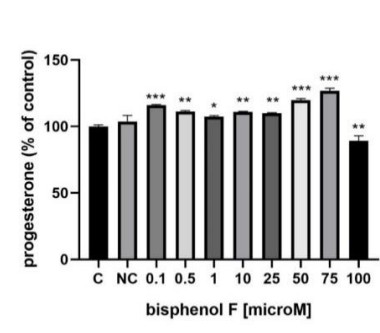
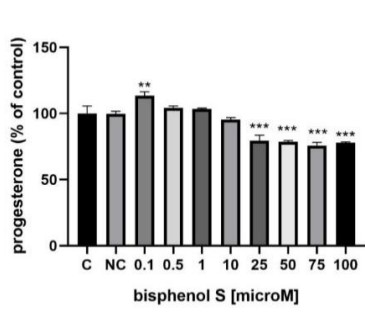

**Figure 5.** The impact of bisphenol B, F, and S on H295R cells progesterone secretion in vitro. Abbreviations: C—control group, NC—negative control. The mean (±SEM) optical density percent of the control (untreated) and treated (bisphenol) groups is represented by each bar. The data were collected from three separate experiments. Between the control and experimental groups, the significance levels were established at *** ($p < 0.001$), ** ($p < 0.01$), and * ($p < 0.05$).

### 3.6. Testosterone Secretion

We observed significant changes in testosterone production after 24 h of cultivation. The effect of bisphenols on the testosterone biosynthesis in H295R cells is presented in Figure 6. When compared with the control group, BPB concentrations of 0.1 μM resulted in a significant increase ($p < 0.01$) in testosterone secretion, whereas the highest dosages (25–100 μM) resulted in a significant decrease ($p < 0.001$) in testosterone secretion. Significantly lower ($p < 0.01$) values were also observed in the experimental groups with the addition of 50, 75, and 100 μM BPF in comparison with a control group, but the lowest concentration (0.1 μM) BPF caused a significant increase ($p < 0.01$) in testosterone secretion by H295R cells. Incubation of H295R cells with BPS for 24 h resulted in significant changes, with increased testosterone production in samples supplemented with 0.1 and 0.5 μM BPS ($p < 0.01$), while higher concentrations of BPS (50, 75, and 100 μM) resulted in significant ($p < 0.01$) testosterone biosynthesis reduction in treated cells.

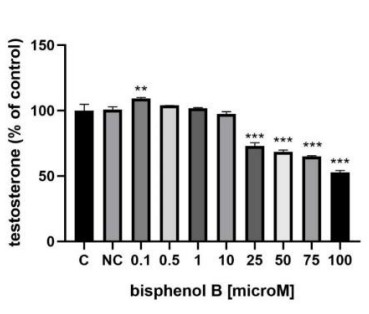
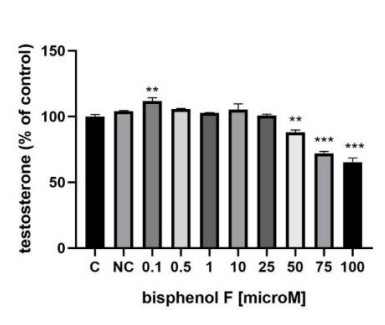
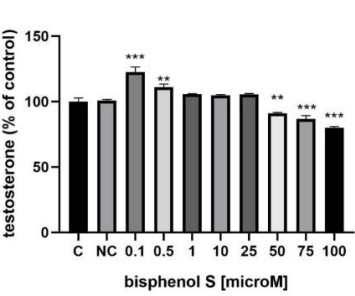

**Figure 6.** The impact of bisphenol B, F, and S on H295R cells testosterone secretion in vitro. Abbreviations: C—control group, NC—negative control. The mean (±SEM) optical density percent of the control (untreated) and treated (bisphenol) groups is represented by each bar. The data were collected from three separate experiments. Between the control and experimental groups, the significance levels were established at *** ($p < 0.001$), ** ($p < 0.01$).

## 4. Discussion

Because of BPA's endocrine-disrupting capabilities and carcinogenic potential, it is progressively being replaced with safer equivalents such as BPB, BPS, and BPF in the manufacturing of polycarbonate plastics and epoxy resins [35]. BPB, BPF, and BPF have all been widely used as BPA substitutes, because of their similar chemical properties, these alternatives may have BPA-like effects. According to environmental monitoring data, these compounds have the potential to become global food contaminants and environmental pollutants in the future. Although these bisphenols are commonly utilized to make a range of daily household items, little is known about their capacity to influence and disrupt steroidogenic pathways, as well as the method by which these chemicals might interfere with steroidogenic enzyme function. Various endocrine disruptors including BPB, BPF, and BPS can have mixed effects on the endocrine system of animals and humans, by acting as direct agonists or antagonists of steroid receptors, as well as potential inducers or inhibitors of steroidogenic enzymes [5,44,45]. The influence of various quantities of BPB, BPF, and BPS on the steroidogenesis of the human H295R cell line is demonstrated in these in vitro experiments (specifically the production of progesterone and testosterone). Furthermore, we assessed the mitochondrial activity and viability parameters (plasma membrane integrity, lysosomal and metabolic activity).

We observed that BPB, BPF, and BPS significantly decreased mitochondrial activity in higher concentrations (50–100 μM). According to the MTT assay, the least toxic was BPS (100 μM = 95.24 ± 0.77%) followed by BPF (100 μM = 92.38 ± 2.01%). BPB showed the highest toxicity (100 μM = 87.92 ± 0.40%). Bisphenol cytotoxicity was shown to increase with increased exposure time and concentration [4]. Our findings are following those of Zhang et al. (2011), who examined the effect of BPA on mitochondrial activity of human H295R cells. Results of this in vitro experiment indicate that BPA at lower concentrations (0.039–10 μM) did not result in any significant inhibition of cell viability [5]. The concentrations of 1 and 10 μM are comparable to the range of blood levels found in persons who had been occupationally exposed [7]. Our findings are supported by another study, which evaluated the cytotoxicity of BPF and BPS on the H295R human adrenocortical carcinoma cell line found that cytotoxicity was not seen at most of the doses examined. Only the highest concentration (100 μM) reduced viability significantly [46]. Feng et al. (2016) used the CCK-8 test to investigate the effects of BPA, BPAF, BPF, and BPS on cell viability. According to their in vitro study, BPAF had the highest cytotoxicity, with 16.1 percent cell viability at 200 μM for 24 h; BPS (62.6 percent cell viability at 200 μM for 24 h) was less toxic than BPA (64.7 percent cell viability at 200 μM for 24 h), and BPF was the least toxic, with no cytotoxicity even at 200 μM [4]. Lan et al. (2017) demonstrated the concentration-dependent biphasic impact of bisphenol A in an in vitro research with MA-10 cells (epithelial-like tumor cell line of Leydig cells from C57BL/6J mice). They treated MA-10 cells treated with various doses (0.01 to 200.0 μM) of BPA. Results showed that low doses (0.01–0.1 μM) of BPA enhanced cell viability, while higher doses (100–200 μM) decreased cell viability [47]. After treatment with high doses of BPS (100 μM) and BPF (100 μM), Huang et al. (2020) found that the viability of KGN cells was significantly reduced [48]. For 24 h, Qi et al. (2020) treated a human ovarian granulosa cell line (KGN) with increasing concentrations of BPA (0.1, 1, 10, and 100 μM). BPA concentrations of 0.1, 1, and 10 μM did not affect the viability of KGN, while 100 μM caused a statistically significant reduction in viability [49]. In our experiments, we treated H295R cells with increasing concentrations (0.1, 0.5, 1, 10, 25, 50, 75, and 100 μM) of BPB, BPF, and BPS, and treatment with all bisphenols with 100 μM resulted in a decrease in viability parameters.

After exposure to 30, 50, and 70 μM BPF (elevated by 733 percent, 1122 percent, and 1273 percent, respectively), Feng et al. (2016) reported increasing levels of progesterone in a dose-dependent manner, while treatment to 1 and 10 μM BPS resulted in significant elevation of progesterone (increased by 50.3 percent and 91.0 percent, respectively) [4]. Their findings are consistent with the findings of our experiments; we observed an increase in free progesterone level after 24 h of BPF exposure for the concentrations 0.1–75 μM.

Only the highest concentration of BPF (100 μM) resulted in a decrease in free progesterone. For 24 h, Qi et al. (2020) treated a KGN cell line with increasing concentrations of BPA (0.1, 1, 10, and 100 μM). KGN's progesterone biosynthesis was significantly reduced after treatment with 10 μM BPA, but during our experiments, we observed a significant increase in testosterone production after treatment with 10 μM BPF and no significant changes after treatment with 10 μM BPB and BPS [49].

There is growing evidence that many chemicals released into the environment can disturb the endocrine system. Certain environmental pollutants can disrupt hormonal balance and interfere with the activities of critical enzymes involved in steroidogenesis, causing or contributing to hormonal disturbance [1]. Goldinger et al. (2015) observed a significant decrease in free testosterone level in an angiotensin-II-responsive steroid-producing adrenocortical cell line culture medium after 48 h of BPS exposure for the concentrations of 1–100 μM [46]. After 24 h of BPS exposure in H295R cell culture media, we observed a decrease in free testosterone levels in culture media for the concentrations 50–100 μM. BPS caused a concentration-dependent reduction in testosterone production at 10–70 μM (reduced by 34.0 percent at 10 μM, 69.1% at 30 μM, 82.4 percent at 50 μM, and 86.8% at 70 μM), according to Feng et al. (2016). However, no significant changes were found in any of the BPF subgroups that were examined [4]. On the other hand, Goldinger et al. (2016) observed an increase in free testosterone level after 48 h of BPF exposure for the concentrations 1–30 μM; only the highest concentration (100 μM) resulted in a decrease in free testosterone, which is in agreement with our results because we observed an increase in free testosterone level after 24 h of BPF exposure for the concentrations 1–30 μM. However, one in three duplicates of their experiment did not match the other two, resulting in a greater standard deviation than expected [46]. Roelofs et al. (2015) observed a significant increase in testosterone concentration in MA-10 Leydig cells after a 48 h-long exposure to BPF at a concentration of 100 μM; however, in our experiment, we observed a significant decrease in testosterone concentration after 24 h exposure to 100 μM of BPF [50].

Endocrine disruptors can disrupt steroid hormone production by inducing or inhibiting enzymatic gene expression via steroid hormone receptors and/or altering enzymatic activity directly or indirectly. Thus, the effect of endocrine disruptors is dependent on the type of issue, pollutant concentrations, bioaccumulation, and stability, as well as the type of exposure (in vitro or in vivo) and duration of exposure. Our findings suggest that testosterone is more sensitive to bisphenol exposure than progesterone. The effect of 17b-hydroxysteroid dehydrogenase's activity is likely to be more sensitive, resulting in lower testosterone release if compared with progesterone. Feng et al. (2016) evaluated transcription levels of corresponding genes encoding steroidogenic enzymes (StAR (Steroidogenic Acute Regulatory Protein), FDX-1 (Ferredoxin 1), CYP11A1 Cytochrome P450 Family 11 Subfamily A Member 1), HSD3B2 (Hydroxy-Delta-5-Steroid Dehydrogenase, 3 Beta- Additionally, Steroid Delta-Isomerase 2), CYP21A2 (Cytochrome P450 Family 21 Subfamily A Member 2), CYP17A1 (Cytochrome P450 Family 17 Subfamily A Member 1), and 17bHSD (17β-Hydroxysteroid dehydrogenases)). The molecular mechanisms by which BPA increases STAR expression are still not fully clarified [4]. In human cumulus granulosa cells, Pogrmic-Majkic et al. (2019) found that increased STAR expression is mediated through the PPAR/EGFR/ERK1/2 (Peroxisome proliferator-activated Receptor, Epidermal Growth Factor Receptor, Extracellular signal-regulated kinases) pathway [51]. Progesterone is an important precursor of hormones such as aldosterone, cortisol, testosterone, and 17b-estradiol in the adrenal glands. It is generated from cholesterol via sequential re-actions catalyzed by StAR, FDX-1, CYP11A1, and HSD3B2. Gene transcription levels of StAR and CYP11A1 were not significantly modified were at any concentrations of BPF and BPS. BPF at concentrations up to 50 μM caused a statistically significant inhibition of HSD3B2 gene expression. However, BPS did not disrupt this gene expression at any tested concentration. Gene transcription involved in testosterone biosynthesis was also measured. Testosterone is generated from progesterone via sequential reactions catalyzed by CYP17A1 and 17b-HSD (17 beta-Hydroxysteroid dehydrogenase). All four compounds

downregulated the gene expression of CYP17A1 in a dose-dependent manner starting at concentrations as low as 1 μM for BPA and BPAF and at concentrations from 10 μM BPF to 30 μM for BPS. However, gene expression of 17b- hydroxysteroid was not interrupted with any of bisphenols across the entire range of concentration, although testosterone production significantly decreased after bisphenol exposure [4]. According to Amar et al. (2020), BPS treatment decreased protein expression of CYP11A1, a steroidogenic enzyme involved in the transformation of cholesterol into pregnenolone, the precursor of progesterone, and therefore a lower pregnenolone concentration might lead to lower progesterone synthesis and release. BPS, on the other hand, did not affect HSD3B1, which is involved in the conversion of pregnenolone to progesterone [52].

Steroid hormone biosynthesis in steroidogenic cells is regulated through trophic hormone activation of protein kinase A (PKA) signaling pathways. Many examples of steroid synthesis regulation via pathways other than the PKA pathway have been identified. In some situations, these pathways work independently of PKA activation, whereas in others, they act synergistically with it. Additional signaling pathways and components include the protein kinase C pathway, arachidonic acid metabolites, growth factors, chloride ion, and the calcium messenger system [53].

Chu et al. (2018) demonstrated that low-dose BPA (1–1000 nM) influenced the ERK signaling pathway and resulted in hormone reductions in human placental cells [54]. Amar et al. (2020) investigated the MAPK3/1 signaling pathway in human granulosa cells in the presence or absence of 10 μM BPS after 5, 10, 30, and 60 min of treatment. After 5 min, a transient 3.7-fold increase in MAPK3/1 phosphorylation was detected in control cells (HGC treated with media alone) ($p$ 0.0001). After 5 min with 10 μM BPS, there was a comparable 4.1-fold increase in MAPK3/1 phosphorylation [52].

## 5. Conclusions

In conclusion, we demonstrated that BPB, BPF, and BPS could affect progesterone and testosterone secretion, as well as affect cell mitochondrial activity, plasma membrane integrity, and lysosomal and metabolic activity. Our research has a few limitations. For the final concentration of BPs, we used data from previous studies. However, the extent to which BPs are taken up by cells is unclear, and our goal was to investigate the possible mechanisms of BPs activity. Another limitation is that the period of exposure in our study was relatively short. Long-term exposure and in vivo experiments may provide more accurate data on their effects.

**Author Contributions:** Conceptualization, N.K. and N.L.; methodology, N.K., H.G., T.J. and K.T.; validation, N.L. and Ł.J.B.; formal analysis, N.K.; investigation, N.K. and N.L.; data curation, N.K. and H.G.; writing—original draft preparation, N.K. and T.J.; writing—review and editing, N.L. and Ł.J.B.; supervision, N.L.; project administration, N.L. and K.T.; funding acquisition, N.L. All authors have read and agreed to the published version of the manuscript.

**Funding:** This research was funded by the Scientific Agency of the Slovak Republic VEGA No. 1/0083/21, No. 1/0038/19, No. 1/0163/18 and by the Slovak Research and Development Agency Grant APVV-20-0218, APVV-19-0243, APVV-18-0312.

**Institutional Review Board Statement:** Not applicable.

**Informed Consent Statement:** Not applicable.

**Data Availability Statement:** All data are provided in the manuscript.

**Conflicts of Interest:** The authors declare no conflict of interest.

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
