# Peer review of "Assessment of the Effective Impact of Bisphenols on Mitochondrial Activity, Viability and Steroidogenesis in a Dose-Dependency in Human Adrenocortical Carcinoma Cells"

_processes, doi:10.3390/pr9081471_

Round 1

Reviewer 1 Report

This is an interesting  and important paper on the endocrine effects of bisphenols B, F and S.  These compounds are hoped to be safe substitutes  of biusphenol A, commonly used for food packaging. The results obtained by the authors show that the use of bisphenols B,F,and S may not be sufficiently safe.The paper merits  publication.

Author Response

We would like to thank the reviewer for their time spent reviewing our manuscript.

Reviewer 2 Report

The manuscript entitled “Assessment of the Effective Impact of Bisphenols on Mitochondrial Activity, Viability and Steroidogenesis in a Dose-Dependency in Human Adrenocortical Carcinoma Cells” aims to elucidate the effects of bisphenol B, F, and S on the steroidogenesis of the human adrenocortical carcinoma cell line. The authors basically describe that, although with the due individual differences between the various compounds, bisphenol B, F, and S increase progesterone and testosterone secretion as well as cell mitochondrial, metabolic, lysosomal activity, and plasma membrane integrity at low doses, with an opposite effect at high doses.  

  1. What are the signaling pathways regulating steroidogenesis induced by low and high doses of bisphenol B, F, and S? The authors should evaluate the main proteins involved such as PKA and PKCs by measuring their total and phosphorylated amount.
  2. Bisphenols are known to induce, in parallel with steroidogenesis, cell growth and migration. The authors should measure these parameters in human adrenocortical carcinoma cell line (H295R) in presence of low and high doses of bisphenol B, F, and S. Futhermore, it could be interesting evaluate ERK1/2 pathway and the expression of matrix metalloproteinases (MMPs) 3 and 9.
  3. Please proofread and fix some typing error throughout the manuscript. Please, focused also on standard English revision.

Author Response

Author's Reply to the Review Report (Reviewer 2):

We would like to thank the reviewer for their thoughtful comments and efforts towards improving our manuscript.

What are the signaling pathways regulating steroidogenesis induced by low and high doses of bisphenol B, F, and S? The authors should evaluate the main proteins involved such as PKA and PKCs by measuring their total and phosphorylated amount.

We've included a section on the signaling pathways that control steroidogenesis at the end of the discussion, however, there is still a scarcity of research in this area.

Thank you for suggesting that we evaluate the total and phosphorylated amounts of the major proteins involved, such as PKA and PKCs. We intended to conduct these analyses as well as protein expression (HSD3B1 and CYP11A1) and gene expression (ESR1, ESR2, ESRRG, GPER, AR, PR, CYP17A1, and STAR) as part of the next phase of our research project, but we need to publish the present results to meet the project's scheduled publication activities.

Bisphenols are known to induce, in parallel with steroidogenesis, cell growth and migration. The authors should measure these parameters in human adrenocortical carcinoma cell line (H295R) in presence of low and high doses of bisphenol B, F, and S. Futhermore, it could be interesting evaluate ERK1/2 pathway and the expression of matrix metalloproteinases (MMPs) 3 and 9.

Thank you for suggesting additional analyses; we planned to conduct cell growth and migration analyses, but we think that it's important to publish the results in phases, and because we sent the article as a "communication" rather than an "article," we hope that the current results will suffice for publication, as the addition of additional analyses and processing would be too time-consuming.

Please proofread and fix some typing error throughout the manuscript. Please, focused also on standard English revision.

We proofread and fixed typing errors throughout the manuscript and revised English using Grammarly.

Reviewer 3 Report

The paper of Knížatová, et al. “Assessment of the Effective Impact of Bisphenols on Mitochondrial Activity, Viability and Steroidogenesis in a Dose-Dependency in Human Adrenocortical Carcinoma Cells” aimed to investigate the effect of bisphenols on the biosynthesis of steroid hormones in human H295R adrenocortical carcinoma cells. I would like to make some remarks that have arisen after reading the manuscript.

Abstract. You have to summarize the article's main findings and indicate the main conclusions or interpretations. In your abstract, only general conclusions were used ("... demonstrated that bisphenols affect progesterone and testosterone secretion, as well as affect cell mitochondrial, lysosomal, and metabolic activity, plasma membrane integrity"). You have to add some basic findings. 

Line 31. What is BPA? (there is no abbreviation description)

Line 41. Is da-ta means data? 

Line 44. "[22] reports ..." Where is the subject?

Line 55. Is mar-keted means marketed? 

Line 62. Wrong writing in "[30]–[33]".

Line 63. "[35] observed ..." Where is the subject?

Line 70 (etc). Wrong writing in "[36], [37]".

Material and Methods. Wrong using of Heading1-3.

Lines 329 (etc). Wrong writing in "50-100 ℘M".

Reference. Wrong description of all references. 

Author Response

We would like to thank the reviewer for their thoughtful comments and efforts towards improving our manuscript.

Abstract. You have to summarize the article's main findings and indicate the main conclusions or interpretations. In your abstract, only general conclusions were used ("... demonstrated that bisphenols affect progesterone and testosterone secretion, as well as affect cell mitochondrial, lysosomal, and metabolic activity, plasma membrane integrity"). You have to add some basic findings.

The abstract in the journal is limited to 200 words; we intended to add specific results in the abstract, however the abstract currently has 200 words. We will correct the abstract in the second round of review if the reviewer still believes it is necessary to modify it.

Line 31. What is BPA? (there is no abbreviation description)

BPA is bisphenol A, we added an abbreviation description

Line 41. Is da-ta means data?

Yes, we made the correction

Line 44. "[22] reports ..." Where is the subject?

We made the correction

Line 55. Is mar-keted means marketed?

Yes, we made the correction

Line 62. Wrong writing in "[30]–[33]".

We corrected, also in lines 58 and 84

Line 63. "[35] observed ..." Where is the subject?

We made the correction

Line 70 (etc). Wrong writing in "[36], [37]".

We corrected writing throughout the text

Material and Methods. Wrong using of Heading1-3.

We made the corrections

Lines 329 (etc). Wrong writing in "50-100 ℘M".

We made the corrections

Reference. Wrong description of all references.

References corrected according to „Instructions for Authors“

Round 2

Reviewer 3 Report

The authors of the manuscript revised all problem moments and the article can be accepted for publication in present form.